# Spontaneous Pneumothorax in COVID-19 Patients Treated with High-Flow Nasal Cannula outside the ICU: A Case Series

**DOI:** 10.3390/ijerph18042191

**Published:** 2021-02-23

**Authors:** Magdalena Nalewajska, Wiktoria Feret, Łukasz Wojczyński, Wojciech Witkiewicz, Magda Wiśniewska, Katarzyna Kotfis

**Affiliations:** 1Department of Nephrology, Transplantology and Internal Medicine, Pomeranian Medical University, 70–111 Szczecin, Poland; nalewajska@gmail.com (M.N.); feretwiktoria@gmail.com (W.F.); wojczynski@gmail.com (Ł.W.); mwisniewska35@gmail.com (M.W.); 2Department of Cardiology, Pomeranian Medical University, 70–111 Szczecin, Poland; witkiewiczwojciech@gmail.com; 3Department of Anesthesiology, Intensive Therapy and Acute Intoxications, Pomeranian Medical University in Szczecin, 70–111 Szczecin, Poland

**Keywords:** coronavirus, COVID-19, HFNC, SARS-CoV-2, pneumothorax

## Abstract

The coronavirus disease 2019 (COVID-19) caused by the severe acute respiratory syndrome coronavirus 2 (SARS-CoV-2) has become a global pandemic and a burden to global health at the turn of 2019 and 2020. No targeted treatment for COVID-19 infection has been identified so far, thus supportive treatment, invasive and non-invasive oxygen support, and corticosteroids remain a common therapy. High-flow nasal cannula (HFNC), a non-invasive oxygen support method, has become a prominent treatment option for respiratory failure during the SARS-CoV-2 pandemic. HFNC reduces the anatomic dead space and increases positive end-expiratory pressure (PEEP), allowing higher concentrations and higher flow of oxygen. Some studies suggest positive effects of HFNC on mortality and avoidance of intubation. Spontaneous pneumothorax has been observed in patients suffering from SARS-CoV-2 pneumonia. Although the viral infection itself contributes to its development, higher PEEP generated by both HFNC and mechanical ventilation is another risk factor for increased alveoli damage and air-leak. Herein, we present three cases of patients with no previous history of lung diseases who were diagnosed with COVID-19 viral pneumonia. All of them were supported with HFNC, and all of them presented spontaneous pneumothorax.

## 1. Introduction

The coronavirus disease 2019 (COVID-19) is a worldwide, rapidly spreading disease, challenging to global healthcare and economics. From March 2020 up to 27 January 2021, there were over 1,475,000 cases reported in Poland alone and nearly 98,281,000 cases globally [1]. A novel coronavirus, named severe acute respiratory syndrome coronavirus 2 (SARS-CoV-2) was identified as an etiological factor, first found in a cluster of patients with pneumonia of unknown origin in a city of Wuhan, China, at the end of 2019 [2].

Typical symptoms of COVID-19 pneumonia are fever, dry cough, and shortness of breath. Other symptoms of SARS-CoV-2 infection may include muscle soreness, fatigue, loss of taste and smell, and gastrointestinal discomfort, including nausea, abdominal pain, diarrhea, or vomiting. Although most of the patients (approx. 80%) have a mild illness and do not require oxygen supplementation or hospitalization, some develop severe pneumonia defined as fever or suspected respiratory infection and one of the following: tachypnoea >30 breaths/min; severe respiratory distress; or oxygen saturation (SpO_2_) ≤ 93% on room air [3]. SARS-CoV-2 pneumonia may be complicated by bacterial coinfection, sepsis, acute respiratory distress syndrome, venous thromboembolism, and even late-onset pulmonary fibrosis [4,5].

There are several classic methods of oxygen supplementation to dyspneic patients – simple nasal cannulas, simple oxygen masks, and oxygen masks with a reservoir, each of them generating a different estimated fraction of inspired oxygen (FiO2), depending on oxygen flow and patient’s respiratory rate. The high-flow nasal cannula (HFNC) is a non-invasive respiratory support machine, designed to deliver 20–80 L/min of air and oxygen mixture that is appropriately heated and humidified and then distributed through wide nasal cannulas. It was previously proved to have a high clinical efficacy in improving ventilation/perfusion ratio, lowering respiratory effort, and raising oxygenation [6,7]. During the COVID-19 pandemic, HFNC gained a lot more attention as a possible way to avoid intubation in some patients [7,8]. Despite HFNC’s overall positive outcomes described in the available literature, we would like to point out its possible adverse effect in SARS-CoV-2 pneumonia—the pneumothorax. Herein, we present three cases of pneumothorax in COVID-19 patients supported on HFNC therapy. We believe that overpressure of the oxygen could be an additional trigger for developing such complications in the presence of SARS-CoV2 induced lung tissue damage.

## 2. Case Reports


*Case 1*


A 64-year-old man with a history of diabetes mellitus type 2, arterial hypertension, and obesity was referred to our department with the diagnosis of severe pneumonia due to SARS-CoV-2 infection. He complained of dyspnea, high-grade fever, and fatigue for 10 days prior to admission. His vital signs at presentation were as follows: a body temperature of 36 °C, blood pressure of 145/80 mmHg, heart rate of 80 beats per minute (BPM), and oxygen saturation of 95% with a mask with a reservoir at 15 L/min oxygen flow. Physical examination was relevant only for bilateral basal crackles. The computed tomography (CT) scan of the lungs displayed diffused bilateral ground-glass opacities. Laboratory findings included elevated C-reactive protein and lactate dehydrogenase levels. All lab results are presented in (Table 1). Community-acquired pneumonia was diagnosed, and standard therapy was initiated with empirical use of ceftriaxone and azithromycin, intravenous dexamethasone, and low molecular weight heparin (LMWH) at prophylactic doses. On day 2, his respiratory function deteriorated, the partial pressure of oxygen to fraction of inspired oxygen (PaO_2_/FiO_2)_ index dropped below 100, and HNFC treatment was initiated at 50–60 L/min and 75–80% FiO_2_ reaching oxygen saturation of 93–94%. Due to further deterioration of respiratory function, the patient was transferred to the intensive care unit (ICU) for continuous care. At admission to ICU, the chest X-ray was performed and showed a 26 mm left-sided pneumothorax (Figure 1), which was further confirmed in the CT scans along with massive subcutaneous emphysema and the suspicion of right-sided pneumothorax (Figure 2). The patient received two doses of iv Tocilizumab 600 mg; the pneumothoraxes were managed conservatively at first, and then two suction drainages were applied in the left pleural cavity. Despite intensive care, the patient died on day 13 of hospitalization.


*Case 2*


An 83-year-old man was admitted to our department due to SARS-CoV-2 related pneumonia with complaints of dyspnea, high-grade fever, headaches, dysgeusia, and dysosmia for five days prior to admission. His past medical history was relevant for arterial hypertension, ischemic heart disease, rheumatoid arthritis, and benign prostatic hyperplasia; he firmly denied smoking history. His vital signs at presentation were body temperature of 36.7 °C, heart rate of 75 BPM, blood pressure of 150/70 mmHg, and oxygen saturation of 90% on atmospheric air. He maintained oxygen saturation of 95% with the supplementation of 15 L/min of oxygen on a mask with a reservoir. Physical examination revealed bilateral crepitations in basal parts of the lungs and the CT scan confirmed the presence of massive bilateral ground-glass opacities corresponding to interstitial pneumonia in the course of COVID-19 infection. His lab results are presented in (Table 1). He was treated in accordance with local guidelines—azithromycin and ceftriaxone were initiated along with intravenous dexamethasone and prophylactic LMWH. On day 7, his oxygen saturation worsened, with the PaO_2_/FiO_2_ index below100 and he was converted to HFNC at 60 L/min and 80–85% FiO_2_, then 80 L/min and 88% FiO_2_. Control chest X-ray displayed a small left-sided pneumothorax (Figure 3) and subcutaneous emphysema on the left side of the neck and chest. The pneumothorax was managed conservatively, however, the patient was qualified for further treatment in the intensive care unit. Unfortunately, the patient died on day 2 of the ICU stay.


*Case 3*


A 56-year-old overweight man with a history of diabetes mellitus type 2 on oral medications and arterial hypertension was admitted to our department due to pneumonia in the course of SARS-CoV-2 infection. He complained of fatigue, loss of appetite, dyspnea, and fever reaching 39 °C for 10 days prior to admission. He has not been previously diagnosed with any lung diseases; he declared no history of smoking. His vital signs at admission were as follows: temperature of 37 °C, blood pressure of 155/100 mmHg, heart rate of 95 BPM, and oxygen saturation of 90% on room air. He was placed on nasal cannulas with the 3 L/min flow of oxygen reaching oxygen saturation of 94%. The physical examination at presentation revealed basal crepitations in the right lung. His laboratory results were relevant for moderately increased inflammatory markers and hyponatremia—all of the results are presented in (Table 1). The CT scan of the chest showed bilateral ground-glass opacities covering less than 50% of the lungs. Standard treatment of community-acquired pneumonia was commenced with azithromycin and ceftriaxone, along with intravenous dexamethasone and prophylactic doses of LMWH. Anti-viral treatment with Remdesivir infusions was also initiated. On day 3, the patient’s respiratory function deteriorated; he was converted to an oxygen mask with a reservoir, and antibiotics were escalated to levofloxacin. Control chest X-ray did not display any abnormalities other than those of interstitial pneumonia. Despite modification of therapy, further worsening of oxygen saturation was observed with the PaO2/FiO2 index below 100, indicating severe acute respiratory distress syndrome (ARDS); therefore, he was qualified for HFNC treatment at 40 L/min and 60% of FiO_2_, temporarily reaching the oxygen saturation of 94%. HFNC was escalated further to 50 L/min and 82% of FiO_2_, however, respiratory failure was still present. The patient was qualified for continuous treatment in the ICU. At admission, another chest X-ray was performed showing a 55-mm pneumothorax in the apex of the left lung (Figure 4). A chest drain was inserted resulting in full resolution of pneumothorax on the next day.

## 3. Discussion

Pneumothorax is the accumulation of air between the visceral and parietal pleura due to active air inflow to the thoracic cavity, as an effect of a loss of intrapleural negative pressure. That causes the lung to collapse partially or totally, which can impair ventilation, oxygenation, or both. The condition varies in its presentation from asymptomatic to life-threatening [9]. Primary spontaneous pneumothorax occurs in generally healthy people without any triggering factor and it is most commonly found in the younger population. Secondary spontaneous pneumothorax is a complication of underlying lung diseases, for example, chronic obstructive pulmonary disease (COPD) or pneumonia.

Spontaneous pneumothorax secondary to lung injury in the course of SARS-CoV-2 has been increasingly reported in the recent literature in the time of the SARS-CoV-2 pandemic. Its incidence in COVID-19 viral pneumonia is estimated to be approximately 0.6–1% of cases [10,11,12]. SARS-CoV-2 infection is responsible for severe pneumonia manifested commonly with acute respiratory distress syndrome (ARDS) leading to the structural changes in the lungs in the form of diffused alveoli wall rupture, and as a consequence, interstitial emphysema [13]. Fibromyxoid exudates, multinucleated giant cells located in the inner space of the alveoli, and the inflammatory cells infiltrates may all also contribute to the alveoli damages observed in SARS-CoV-2 pneumonia [11]. Other typical findings associated with ARDS, the hyaline membranes, and desquamation of pneumocytes could also be present in biopsy specimens [14]. 

Despite the viral infection itself, the overdistention of the alveoli detected during both invasive and non-invasive ventilatory support is an additional risk factor for developing a pneumothorax. The underlying mechanism is thought to be related to barotrauma, which is a trauma caused by increased transalveolar pressure, leading to alveoli rupture. Barotrauma is more frequently observed in patients ventilated mechanically than those ventilated non-invasively. The incidence of barotrauma events, namely, pneumothoraces and pneumomediastinum in mechanically ventilated patients suffering from COVID-19, was reported to be approximately 15% [15]. Its increased incidence is related to the presence of high peak inspiratory pressures (PIP), high positive end-expiratory pressure (PEEP), high tidal volumes and minute ventilation, and prolonged ARDS, in addition to the manipulation regarding the airway (intubation or tracheostomy) [10,16].

Some authors also connect persistent cough, a common symptom of COVID-19, with an increased risk of spontaneous pneumothorax due to high intrapulmonary strain [16,17]. Coughing causes a rapid increase in intra-pulmonary pressure—while the glottis is closed and intracostal muscles contract together with the diaphragm, the intrathoracic pressure can rise up to 300 mmHg (circa 400 cm water pressure) [18].

HNFCs are proved to be useful in specific groups of patients. This refers to those with acute hypoxemic respiratory failure, post-surgical respiratory failure, chronic obstructive pulmonary disease exacerbation, pulmonary edema due to acute heart failure, or obstructive sleep apnea. There are several physiological mechanisms that contribute to its efficiency—helping to wash out carbon dioxide by recruiting physiological dead space in the alveoli, decreasing respiratory rate, increasing tidal volume and end-expiratory volume, and last but not least, generating a PEEP thus preventing the alveoli from collapsing [6,19]. Approximately, in perfect conditions, i.e., when a patient has his mouth closed during HFNC ventilation, PEEP raises about 1 cm water pressure with every 10 liters of flow [19]. That means that in our described above, we could generate a PEEP of 6 cm water pressure. We, therefore, wonder if diffuse alveolar damage observed in severe COVID-19 could be making the patients more prone to barotrauma, even though the peak expiratory pressure during HFNC is not as high as during invasive mechanical ventilation on the ICU. Of course, the authors are aware that susceptibility to pneumothorax is highly dependent on the status of lung tissue and that the usage of HFNC cannot be seen as a standalone risk factor. In our case series, pneumothoraces occurred in patients receiving HFNC oxygen therapy, but it could also be that the use of HFNC was just a surrogate for the severity of the patients’ condition. Flourishing cytokine storm, bacterial co-infection and increased respiratory effort and coughing most likely play an important role in the pathomechanism of this rare complication. What is worth noting is that in our department, we did not observe pneumothoraxes in COVID-19 patients that were ventilated with low-flow oxygen, and we hospitalized more than 150 people since August 2020. Nevertheless, it has been our priority to engage all possible means of treatment before transferring the patient to an intensive care unit to prevent intubation and mechanical ventilation in COVID-19 patients [20,21]. 

We would like to raise two main concerns about using HFNC in COVID-19 patients. First, clinicians should be aware that pneumothorax might be more frequent in the group of patients requiring high-flow oxygen. Therefore, rapid deterioration in oxygenation might indicate this complication and should always entail differential diagnosis. Second, we claim that HFNC might not be as safe as presumed, especially in patients with COVID-induced diffuse alveolar destruction. Hence, we would next time consider putting a patient onto HFNC before the alveolar destruction exceeds, and PaO2/FiO2 ratio drops below 100. 

## 4. Conclusions

We would like to draw attention to a rare but serious complication of SARS-CoV-2 pneumonia—pneumothorax. Due to the rapid spread of the COVID-19 pandemic, many patients require support with high-flow oxygenation. We believe that clinicians should be aware that HFNC may be associated with a higher incidence of barotrauma in comparison to the standard, low-flow therapies. We presume that due to generating PEEP, it may additionally contribute to alveolar damage by rupturing their walls in the presence of diffuse neutrophilic infiltrations in the alveoli. On the other hand, it surely is safer than mechanical ventilation, because the PEEP generated using high-flow nasal cannulas is not as high as in closed-system devices. Every case of the rapid deterioration of oxygenation in a patient supported with HFNC should raise a suspicion of pneumothorax and lead to proper further decision making. Nevertheless, HFNC is a relatively safe and very often a superb method of ventilation for many patients with SARS-CoC-2 induced respiratory failure.

## Figures and Tables

**Figure 1 ijerph-18-02191-f001:**
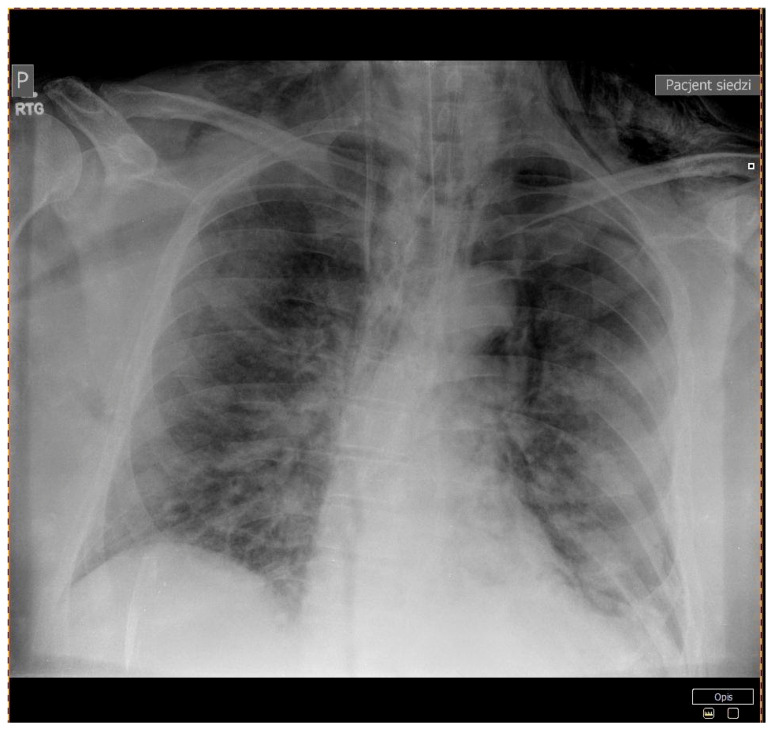
Patient 1 - A chest X-ray showing a 26 mm left-sided pneumothorax.

**Figure 2 ijerph-18-02191-f002:**
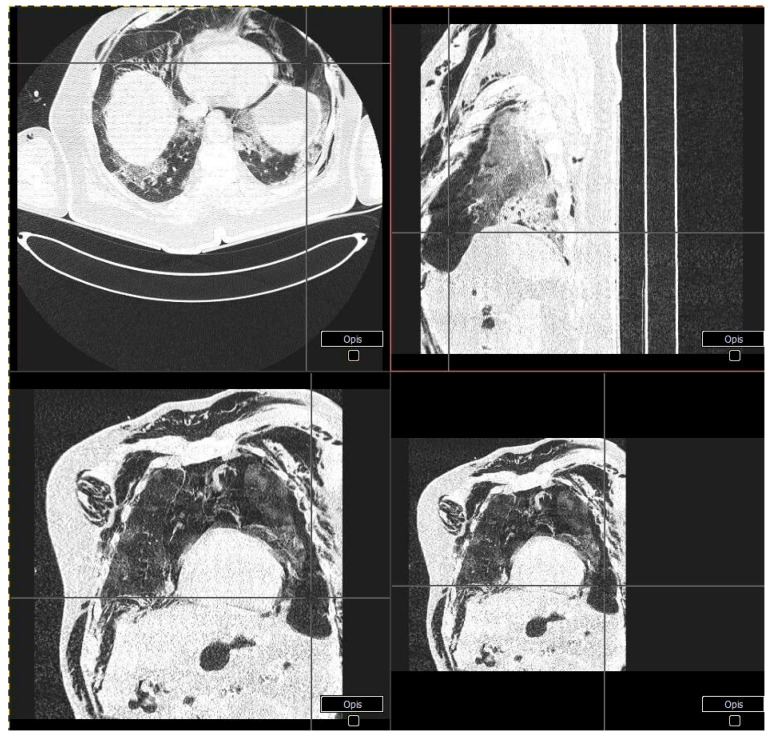
Patient 1 - CT scans showing left-sided pneumothorax along with massive subcutaneous emphysema and the suspicion of right-sided pneumothorax.

**Figure 3 ijerph-18-02191-f003:**
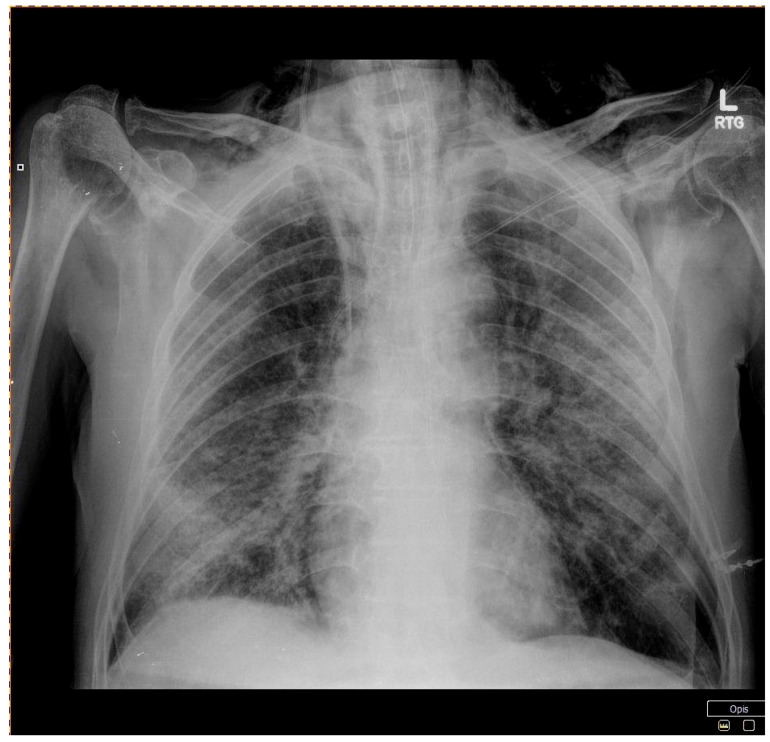
Patient 2 - A chest X-ray showing a small left-sided pneumothorax.

**Figure 4 ijerph-18-02191-f004:**
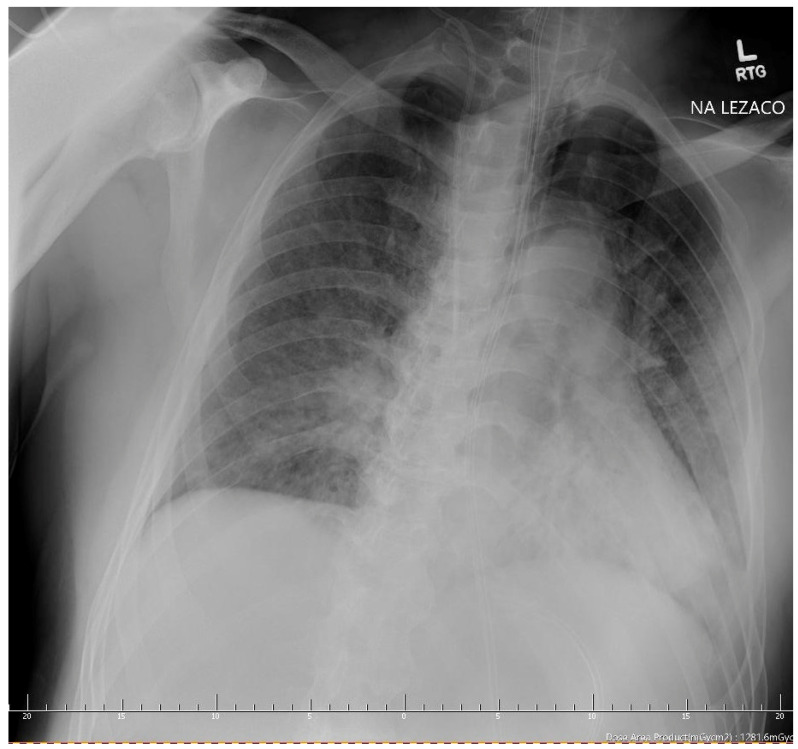
Patient 3 - A chest X-ray showing left-sided pneumothorax.

**Table 1 ijerph-18-02191-t001:** Patient characteristics.

Patient Characteristics (Sex, Age)		Patient 1 (Male, 64)		Patient 2 (Male, 83)		Patient 3 (Male, 56)	
		At Admission to the Hospital	1 Day After Conversion of Respiratory Support	At Admission to The Hospital	1 Day After Conversion of Respiratory Support	At Admission to the Hospital	1 Day After Conversion of Respiratory Support
**Co-morbidities**		Arterial hypertension, diabetes mellitus type 2		Rheumatoid arthritis, arterial hypertension, benign prostate hyperplasia, ischemic heart disease		Diabetes mellitus type 2, arterial hypertension	
**BMI [kg/m^2^]**		37		21.6		29.22	
**Obesity class**		Class 2 Obesity		Normal		Overweight	
**Symptoms**		Dyspnea, fever, weakness		Fever, headache, dyspnea, taste and smell disturbances		Fatigue, dyspnea, fever, loss of appetite	
**CT scan**		Patchy and partially coalescing geographic ground-glass opacities		Diffuse bilateral ground-glass opacities and fibrotic changes In the form of bronchiectasis in lower lobes		Bilateral ground-glass opacities covering less than 50% of the lungs	
**Respiratory support**	**Type**	Non-rebreathing facemask	High-flow nasal cannula	Non-rebreathing facemask	High-flow nasal cannula	Simple nasal cannula	High-flow nasal cannula
	**Flow [L/min]**	14	50	15	60	3	50
	**FiO_2_ (%)**	>90	75	>90	80	>90	82
	**Temperature**	~24 °C (room temp.)	31 °C	~24 °C (room temp.)	31 °C	~24 °C (room temp.)	31 °C
**Saturation (%)**		94	90	95	98	94	71
**pH**		7.40	7.49	7.49	7.53	7.57	7.51
**pO_2_ (mmHg)**		73	54	78	96	66	33
**pCO_2_ (mmHg)**		41	33	30	33	28	28
**Lactate (mmol/L)**		1.1	1.4	1.3	1.6	1.6	1.9
**WBC (G/L)**		9.9	16.3	5.3	16.3	8.33	13.5
**Hgb (mmol/L)**		8.9	8.3	8.1	7.8	9.9	8.5
**IL-6 (pg/mL)**		10.2	202.0	40.8	439.0	110	128
**D-dimer (ng/mL)**		1026	1566	4820	>7650	700	2206
**LDH (U/L)**		523.0	523.0	592.0	-	-	771
**CRP (mg/L)**		81.7	94.6	64.5	40.9	29	161
**Serum creatinine (mg/dL)**		1.0	0.8	0.8	0.7	1.18	0.78
**Serum sodium (mmol/L)**		139	139	136	139	129	128
**Serum potassium (mmol/L)**		4.8	5.3	3.9	4.3	4.24	5.0

Legend: BMI-body mass index; FiO_2_ -fraction of inspired oxygen; pO_2_ -partial pressure of oxygen; pCO_2_ -partial pressure of carbon dioxide; WBC-white blood cells count; Hgb-hemoglobin; Il-6-Interleukin-6; LDH-lactate dehydrogenase; CRP-C-reactive protein; CT – computed tomography.

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
