# Peer review of "Spontaneous Pneumothorax in COVID-19 Patients Treated with High-Flow Nasal Cannula outside the ICU: A Case Series"

_ijerph, 2021, doi:10.3390/ijerph18042191_

Round 1

Reviewer 1 Report

The authors described 3 cases of spontaneous pneumothorax by Covid pneumonia patients. 

I am not sure, that the pneumothorax definition lika the accumulation of air between thoracic pleura and lung pleura is sufficient now. The patogenesis of that depend from active sucction of the air in the thoracic cavity due the communication of pleural cavity with outside gas pressure and loosing the negative air pressure in the pleural cavity. The pneumothorax associate with colapse of the lung (one half). 

The last sentence in the introduction can be improve. 

In disscusion we can´t find the some data about the presence of pneumothorax by Covid 19 positive pneumonia patients with tracheostomy.  This is more important for the decision of pathogenesis of pneumothoray. The pneumothorax can depend by lung tissue status (inlammation, necrosis a.o.) and/or the overpressure of the oxygen (gases) during the PEEP procedure by closed respiratory system (intubation, tracheostomy). The barotrauma of the lung for nasal canula (the open system) is reducing of the PEEP (PIP) by no-closed system.  

This eventuality was not disscused, that we can be not sure that the all facts were disscused for explanation of the rare complication.  

Author Response

Dear Reviewer,

thank you very much for your kind remarks.

1. Following your guidance we modified the definition of pneumothorax to (line 154-158): "Pneumothorax is the accumulation of air between the visceral and parietal pleura due to active air inflow to the thoracic cavity, as an effect of a loss of intrapleural negative pressure. That causes the lung to collapse partially or totally, which can impair ventilation, oxygenation or both. The condition varies in its presentation from asymptomatic to life-threatening [9]." and added one more reference: Zarogoulidis P, Kioumis I, Pitsiou G, et al. Pneumothorax: from definition to diagnosis and treatment. J Thorac Dis. 2014;6(Suppl 4):S372-S376. doi:10.3978/j.issn.2072-1439.2014.09.24

2. We modified the last sentence in the introduction to (line 62-64): "Herein, we present three cases of pneumothorax in COVID-19 patients supported on HFNC therapy. We believe that overpressure of the oxygen could be an additional trigger for developing such complication in the presence of SARS-CoV2 induced lung tissue damage."

3. In response to the following sentence "In disscusion we can´t find the some data about the presence of pneumothorax by Covid 19 positive pneumonia patients with tracheostomy." we would like to explain that none of the patients has a tracheostomy in situ. However, to elaborate on the pathogenesis of pneumothorax we added the following information (line 180-189): "The incidence of barotrauma events, namely pneumothoraces and pneumomediastinum in mechanically ventilated patients suffering from COVID-19 was reported to be approximately 15% [15]. Its increased incidence is related to the presence of high peak inspiratory pressures (PIP), high positive end-expiratory pressure (PEEP), high tidal volumes and minute ventilation, as well as prolonged ARDS, but also to the manipulation reagrding the airway (intubation or trachestomy) [10,16]Some authors also connect persistent cough, a common symptom of COVID-19, with increased risk of spontaneous pneumothorax due to high intrapulmonary strain [16]. Coughing causes rapid increase in intra-pulmonary pressure: while glottis is closed and intracostal muscles contract together with the diaphragm, the intrathoracic pressure can rise up to 300 mmHg (circa 400 cm water pressure)[18]."

And also more information (line 200-206) "We therefore wonder if diffuse alveolar damage observed in severe COVID-19 could be making the patients more prone to barotrauma, even though the peak expiratory pressure during HFNC is not as high as during invasive mechanical ventilation on the ICU. Of course, the authors are aware that susceptibility to pneumothorax is highly dependent on the status of lung tissue and that the usage of HFNC cannot be seen as a standalone risk factor. Flourishing cytokine storm, bacterial co-infection and increased respiratory effort as well as coughing most likely play an important role in the pathomechanism of this rare complication."

We hope that this improvement is satisfactory to you.

With best regards

Katarzyna Kotfis, MD, PhD, EDAIC, Associate Professor

Reviewer 2 Report

The authors present an interesting observation in sick Covid-19 patients, namely spontaneous pneumothorax. The causality between HFNC and pneumothorax is less clear and I think claiming that pneumothorax is due to the use of HFNC is not justified. Certainly they occurred in patients receiving HFNC oxygen therapy, but it could be that the use of HFNC was just a surrogate for the severity of the patients' condition. The paper would be much-enhanced if the authors also explained the possible role of high intra-pleural pressure gradients due to coughing, straining and respiratory effort (due to severe dyspnea) as possible contributors to the formation of spontaneous pneumothoraces.

Author Response

Dear Reviewer, 

thank you very much for your kind remarks. 

Following your guidance we added the following information to the discussion:

1. Line 205-207: "In our case series pneumothoraces occurred in patients receiving HFNC oxygen therapy, but it could be that the use of HFNC was just a surrogate for the severity of the patients' condition."

2. To explain the possible role of high intra-pleural pressure gradients due to coughing, straining and respiratory effort (due to severe dyspnea) as possible contributors to the formation of spontaneous pneumothoraces we added the following information (line 180-190): 

"The incidence of barotrauma events, namely pneumothoraces and pneumomediastinum in mechanically ventilated patients suffering from COVID-19 was reported to be approximately 15% [15]. Its increased incidence is related to the presence of high peak inspiratory pressures (PIP), high positive end-expiratory pressure (PEEP), high tidal volumes and minute ventilation, as well as prolonged ARDS, but also to the manipulation reagrding the airway (intubation or trachestomy) [10,16]

Some authors also connect persistent cough, a common symptom of COVID-19, with increased risk of spontaneous pneumothorax due to high intrapulmonary strain [16]. Coughing causes rapid increase in intra-pulmonary pressure: while glottis is closed and intracostal muscles contract together with the diaphragm, the intrathoracic pressure can rise up to 300 mmHg (circa 400 cm water pressure) [18]."      

We certainly hope that we have addressed all your remarks sufficiently.

With best regards

Katarzyna Kotfis, MD, PhD, EDAIC, Associate Professor 

Round 2

Reviewer 2 Report

The authors have addressed the concerns and comments raised in my original review.

The paper could still benefit from English sub-editing.